# Mapping Seasonal Leaf Nutrients of Mangrove with Sentinel-2 Images and XGBoost Method

**Jing Miao [1,2], Jianing Zhen [3], Junjie Wang [1,4,***](image), Demei Zhao [1,2], Xiapeng Jiang [1,2], Zhen Shen [1,2], Changjun Gao [5] and Guofeng Wu [1,2]**

1   MNR Key Laboratory for Geo-Environmental Monitoring of Great Bay Area, Guangdong Key Laboratory of Urban Informatics, Shenzhen Key Laboratory of Spatial Smart Sensing and Services, Shenzhen University, Shenzhen 518060, China; 2060325008@email.szu.edu.cn (J.M.); 2050322003@email.szu.edu.cn (D.Z.); 1900325011@email.szu.edu.cn (X.J.); 2100325026@email.szu.edu.cn (Z.S.); guofeng.wu@szu.edu.cn (G.W.)
2   School of Architecture and Urban Planning, Shenzhen University, Shenzhen 518060, China
3   Guangdong Laboratory of Artificial Intelligence and Digital Economy (SZ), Shenzhen 518000, China; zhenjn@radi.ac.cn
4   College of Life Sciences and Oceanography, Shenzhen University, Shenzhen 518060, China
5   Guangdong Provincial Key Laboratory of Silviculture, Protection and Utilization, Guangdong Academy of Forestry, Guangzhou 510520, China; gaochangjun015@163.com
*   Correspondence: wang_2015@szu.edu.cn

**Abstract:** Monitoring the seasonal leaf nutrients of mangrove forests helps one to understand the dynamics of carbon (C) sequestration and to diagnose the availability and limitation of nitrogen (N) and phosphorus (P). To date, very little attention has been paid to mapping the seasonal leaf C, N, and P of mangrove forests with remote sensing techniques. Based on Sentinel-2 images taken in spring, summer, and winter, this study aimed to compare three machine learning models (XGBoost, extreme gradient boosting; RF, random forest; LightGBM, light gradient boosting machine) in estimating the three leaf nutrients and further to apply the best-performing model to map the leaf nutrients of 15 seasons from 2017 to 2021. The results showed that there were significant differences in leaf nutrients ($p < 0.05$) across the three seasons. Among the three machine learning models, XGBoost with sensitive spectral features of Sentinel-2 images was optimal for estimating the leaf C ($R^2$ = 0.655, 0.799, and 0.829 in spring, summer, and winter, respectively), N ($R^2$ = 0.668, 0.743, and 0.704) and P ($R^2$ = 0.539, 0.622, and 0.596) over the three seasons. Moreover, the red-edge (especially B6) and near-infrared bands (B8 and B8a) of Sentinel-2 images were efficient estimators of mangrove leaf nutrients. The information of species, elevation, and canopy structure (leaf area index [LAI] and canopy height) would be incorporated into the present model to improve the model accuracy and transferability in future studies.

**Keywords:** mangrove; Sentinel-2 image; seasonal leaf nutrients; XGBoost; red edge





## 1. Introduction

Carbon (C) is the most abundant nutrient element in the dry matter of leaves [1], and nitrogen (N) and phosphorus (P) are essential nutrients in the construction of nucleic acid and proteins in plants [2]. Mangrove forest is one of the most species-diverse and productive marine ecosystems [3], which are the main contributors to blue C in coastal zones [4]. Leaf nutrients often vary between seasons to adapt to the growth process and seasonal climatic cycles [5]. Seasonal nutrient monitoring helps to reflect the dynamics of C sequestration and diagnose the availability and limitation of N and P, which is important to understanding the growth status and nutrient utilization strategies of the mangrove ecosystem [6].

In the last two decades, airborne and satellite-based remote sensing techniques have been key methods in the dynamic monitoring of mangrove growth and health [7]. At

the landscape scale, most studies have focused on the mapping of the extent, species composition, biomass, leaf area index (LAI), and chlorophyll content of mangrove forests using medium- (e.g., Landsat-7/8 and Sentinel-2) and high-spatial-resolution satellite (e.g., Worldview-2/3 and Pléiades-1) and Unmanned Aerial Vehicle (UAV-based) multispectral and hyperspectral images [8–10]. In the visible and near-infrared bands, vegetation has obvious reflection and absorption features which are formed by the electronic transition of electromagnetic radiation, vibration, and harmonic generation of chemical bonds (e.g., C-H and N-H bond) through the chemical components of plant leaves [11]. Based on the spectral mechanism, leaf nutrients (C, N, and P) are widely estimated in crop and grassland ecosystems [12,13], however, such estimations have rarely been investigated in mangrove forests.

Due to the abundant spectral details, proximal and UAV-based hyperspectral data are mainly used in the remote estimation of leaf nutrients, and several studies claimed that red-edge wavelengths are sensitive to leaf N and P estimation [13]. Satellite-based hyperspectral imagery is rarely used, because there are far fewer satellites in orbit than multispectral satellites. Among the commonly used multispectral satellites, only Sentinel-2 contains three red-edge bands sensitive to plant growth and health, which could provide great potential for mapping mangrove seasonal leaf nutrients. Yet, little attention has been paid to the long-term monitoring of mangrove seasonal leaf nutrients using Sentinel-2 imagery.

Since leaf C, N, and P are not input parameters to physical models, these nutrients have only been directly estimated with empirical models, including simple regression with one vegetation index [14,15], multiple linear regression (e.g., partial least squares regression) [16,17], and machine learning models (e.g., random forest [RF] and support vector machine [SVM]) [18]. Moreover, the machine learning model generally obtains higher accuracy in estimating leaf nutrients [19]. Recently, due to good accuracy and rapid computational speed, gradient boosting algorithms (e.g., XGBoost [extreme gradient boosting] and LightGBM [light gradient boosting machine]) have been successfully used to remotely assess the parameters of ecological environments (e.g., biomass, soil organic carbon, and leaf chlorophyll content) [18,20,21]. However, the performances of the XGBoost and LightGBM models have not been evaluated in mapping mangrove leaf nutrients.

Using seasonal Sentinel-2 images of mangrove forests in Gaoqiao Mangrove Reserve, China, this study aimed to explore the seasonal response of mangrove leaf nutrients (C, N, and P), compare three machine learning models (XGBoost, RF, and LightGBM) in estimating leaf nutrients, and further to extend the best-performing model to map the leaf nutrients of 15 seasons from 2017 to 2021. The results could facilitate our understanding of seasonal nutrient cycling and limitations in mangrove ecosystems.

## 2. Materials and Methods

### 2.1. Field Sampling

Three field surveys were conducted in three seasons (spring of 2018, winter of 2019, and summer of 2020) in Gaoqiao Mangrove Reserve (Figure 1), Guangdong Province, China. The dominant mangrove species of the reserve are *Aegiceras corniculatum*, *Bruguiear gymnorrhiza*, *Avicennia marina*, *Rhizophora stylosa*, *Sonneratia apetala*, and *Kandelia candel*. With a random sampling strategy and field survey accessibility, a total of 53, 62, and 57 plots (15 m × 15 m) containing one single species (Table 1) were randomly set in 2018, 2019, and 2020, respectively. For each plot, five trees were randomly selected with a distance of 2–5 m between trees, and five mature and healthy leaves were randomly collected from the top canopy of each tree. The geographical location of the center of each plot was recorded by a differential GPS with a positional accuracy less than 20 cm to avoid position mismatch between the image and plot, hence, the corresponding pixel (10 m × 10 m) of geometrically corrected Sentinel-2 imagery could represent the spectral information of the plot.

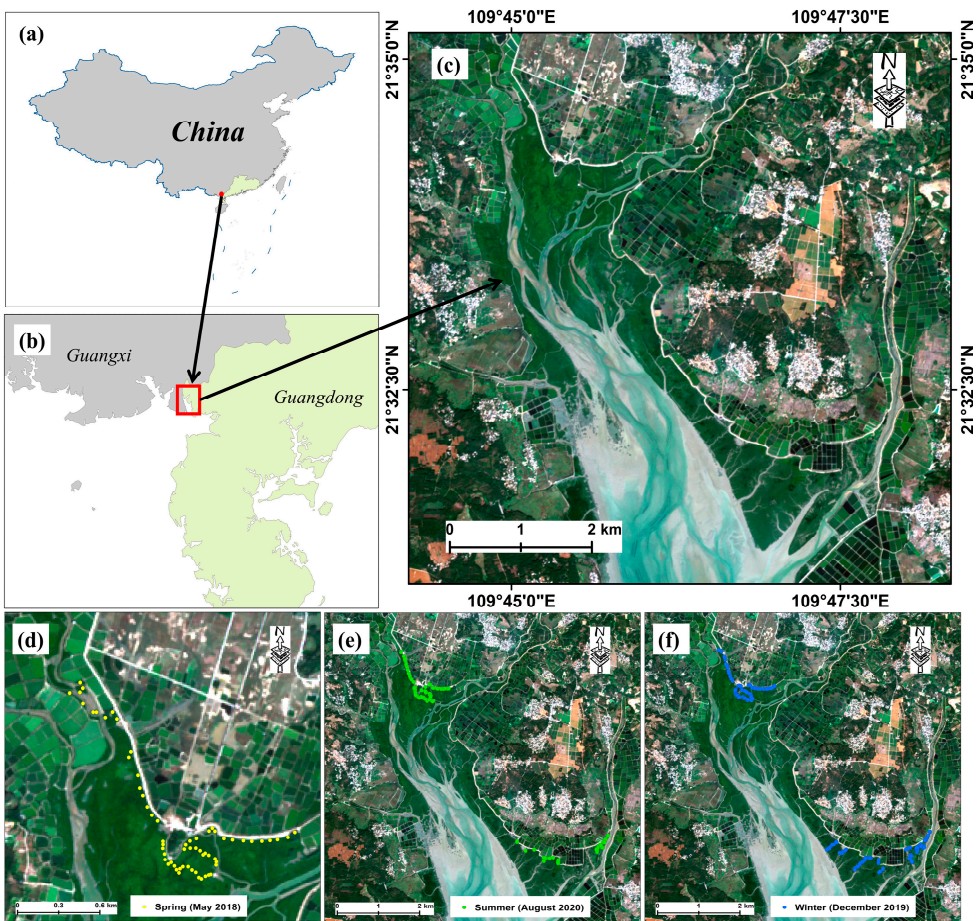

**Figure 1.** Study area (**a**–**c**) and spatial distribution of sampling plots in three seasons (**d**–**f**) (spring: 22–26 May 2018; winter: 20–28 December 2019; summer: 6–12 August 2020). The true-color image is based on the Sentinel-2 image (date: 24 November 2019) with a color combination of band 4 (red), band 3 (green), and band 2 (blue).

**Table 1.** Number of plot samples collected in three seasons.

| Species \ Season | Spring in 2018 | Summer in 2020 | Winter in 2019 |
|---|---|---|---|
| *Aegiceras corniculatum* | 35 | 16 | 23 |
| *Bruguiear gymnorrhiza* | 10 | 13 | 22 |
| *Avicennia marina* | 0 | 16 | 6 |
| *Rhizophora stylosa* | 3 | 6 | 8 |
| *Sonneratia apetala* | 1 | 3 | 0 |
| *Kandelia candel* | 4 | 3 | 3 |

## 2.2. Chemical Analysis of Leaf Nutrients

All the fresh leaf samples were dried in an oven at 65 °C for 72 h. Afterwards, C, N, and P concentrations (unit: g/kg) were determined (Table 2) by the high temperature external thermal potassium dichromate oxidation-volumetric method [22], Kjeldahl method including digestion, distillation, and titration [23], and vanadate–molybdate yellow colorimetric method using sulfuric acid–hydrogen peroxide digestion [24], respectively.

**Table 2.** Statistics of leaf nutrients (unit: g/kg) in three seasons.

| Season (Number of Samples) | Nutrient | Min | Max | Mean | CV (%) |
|---|---|---|---|---|---|
| Spring (53) | C | 408.38 | 481.02 | 449.07 | 3.54 |
| | N | 8.05 | 15.61 | 10.58 | 15.49 |
| | P | 0.70 | 1.85 | 0.90 | 19.21 |
| Summer (62) | C | 424.73 | 542.08 | 492.75 | 5.93 |
| | N | 7.23 | 19.28 | 10.59 | 22.64 |
| | P | 0.62 | 2.00 | 0.86 | 28.24 |
| Winter (57) | C | 403.48 | 501.87 | 456.66 | 6.49 |
| | N | 9.77 | 22.86 | 14.94 | 20.98 |
| | P | 0.83 | 2.68 | 1.57 | 33.99 |

CV: coefficient of variation.

### 2.3. Sentinel-2 Images Pre-Processing and Vegetation Indices Extraction

Three cloudless Sentinel-2 images (Level 1C, acquisition date: 23 May 2018, 24 November 2019, and 23 August 2020), acquired as consistently with the sampling date as possible, were downloaded from the USGS official website (https://earthexplorer.usgs.gov/, accessed on 20 July 2021). The Sen2Cor module from the Sentinel Application Platform (SNAP) was used for atmospheric correction, converting the image top-level reflectance to canopy reflectance (12 spectral bands in the range of 443–2190 nm, Table 3). The bands (B5, B6, B7, B8, B11, and B12) with a spatial resolution of 20 m and 60 m were then resampled to 10 m using the Sen2Res module in SNAP (http://step.esa.int/main/snap-supported-plugins/sen2res/, accessed on 20 July 2021). The Sen2Res algorithm was proposed by Brodu [25], and it employs the general geometric information to unmix low-resolution pixels to realize super-resolution reconstruction of low-resolution bands while keeping the spectral characteristics unchanged.

**Table 3.** Band information of the Sentinel-2 satellite.

| Band | Center Wavelength/nm | Bandwidth/nm | Spatial Resolution/m |
|---|---|---|---|
| B1 (Coastal aerosol) | 443 | 20 | 60 |
| B2 (Blue) | 490 | 65 | 10 |
| B3 (Green) | 560 | 35 | 10 |
| B4 (Red) | 665 | 30 | 10 |
| B5 (Red-edge1) | 705 | 15 | 20 |
| B6 (Red-edge2) | 740 | 15 | 20 |
| B7 (Red-edge3) | 783 | 20 | 20 |
| B8 (NIR) | 842 | 115 | 10 |
| B8a (Narrow NIR) | 865 | 20 | 20 |
| B9 (Water Vapor) | 945 | 20 | 60 |
| B10 (Cirrus) | 1380 | 30 | 60 |
| B11 (SWIR1) | 1610 | 90 | 20 |
| B12 (SWIR2) | 2190 | 180 | 20 |

With the 12 bands of 10 m spatial resolution, a total of 30 vegetation indices (VIs, Table 4) were extracted for each image. Some studies claimed that leaf N and P had a close relationship with leaf pigments [12,26], and red-edge bands were sensitive to leaf N and P [13], hence, 28 selected VIs were related to leaf chlorophyll, carotenoid, anthocyanin, and red-edge bands. Moreover, two mangrove-related VIs (MI and MFI) were chosen. The Pearson's correlations of mangrove leaf C, N, and P concentrations against the spectral features of Sentinel-2 images (12 bands + 30 VIs) were calculated and compared for the three seasons. The images were then geometrically corrected with a UAV-based digital orthophoto (spatial resolution: 0.2 m).

**Table 4.** Vegetation indices used in this study.

| Vegetation Index | Formula | Sentinel-2 Bands | Reference |
|---|---|---|---|
| Normalized Difference Vegetation Index (NDVI) | $(R_{NIR} - R_{red})/(R_{NIR} + R_{red})$ | B5, B8 | [27] |
| Green NDVI (gNDVI) | $(R_{750} - R_{550})/(R_{750} + R_{550})$ | B3, B6 | |
| Optimized Soil-Adjusted Vegetation Index (OSAVI) | $1.16 * (R_{800} - R_{670})/(R_{800} + R_{670} + 0.16)$ | B4, B8 | [28] |
| Red-Edge Inflection Point (REIP) | $R_{700} + 40 * \frac{0.5*(R_{670} - R_{780}) - R_{700}}{R_{740} - R_{700}}$ | B4, B5, B6, B7 | [29] |
| Simple Ratio Index (SR$_{705}$) | $R_{750}/R_{705}$ | B5, B6 | [30] |
| Enhanced Vegetation Index (EVI) | $2.5 * (R_{NIR} - R_{red})/(R_{NIR} + 6 * R_{red} - 7.5 * R_{blue} + 1)$ | B2, B4, B8 | [31] |
| SR$_{Chl\,a}$ | $R_{672}/(R_{550} * R_{708})$ | B3, B4, B5 | |
| SR$_{Chl\,b}$ | $R_{672}/R_{550}$ | B3, B4 | [27] |
| SR$_{chl}$ | $R_{860}/(R_{550} * R_{708})$ | B3, B5, B8 | |
| Modified Cab Absorption in Reflectance Index (MCARI) | $[(R_{700} - R_{670}) - 0.2 * (R_{700} - R_{550})] * (R_{700}/R_{670})$ | B3, B4, B5 | [32] |
| Modified Chlorophyll Absorption in Reflectance Index (MCARI1) | $1.2 * [2.5 * (R_{800} - R_{670}) - 1.3 * (R_{800} - R_{550})]$ | B3, B4, B8 | |
| Transformed CARI (TCARI) | $3 * [(R_{700} - R_{670}) - 0.2 * (R_{700} - R_{550}) * (R_{700}/R_{670})]$ | B3, B4, B5 | [33] |
| MCARI/OSAVI | $\frac{[(R_{700} - R_{670}) - 0.2*(R_{700} - R_{550})]*(R_{700}/R_{670})}{[1.16*(R_{800} - R_{670})/(R_{800} + R_{670} + 0.16)]}$ | B3, B4, B5, B8 | [34] |
| TCARI/OSAVI | $\frac{3*[(R_{700} - R_{670}) - 0.2*(R_{700} - R_{550})]*(R_{700}/R_{670})}{[1.16*(R_{800} - R_{670})/(R_{800} + R_{670} + 0.16)]}$ | B3, B4, B5, B8 | |
| Red-Edge Position (REP) | $705 + 35 * [0.5 * (R_{665} + R_{783}) - R_{705}]/(R_{740} - R_{705})$ | B4, B5, B6, B7 | [35] |
| Pigment Specific Simple Ratio for Chla (PSSRa) | $R_{800}/R_{675}$ | B4, B8 | [36] |
| Green chlorophyll index (CI$_{green}$) | $(R_{RE3}/R_{green}) - 1$ | B3, B7 | [37] |
| Green chlorophyll index (CI$_{red\text{-}edge}$) | $(R_{RE3}/R_{red\,edge}) - 1$ | B5, B7 | |
| Disease Water Stress Index (DSWI) | $(R_{803} + R_{549})/(R_{1659} + R_{681})$ | B3, B4, B8, B11 | [38] |
| Moisture Stress Index (MSI) | $R_{SWIR}/R_{NIR}$ | B8, B11 | [39] |
| Red and Green Pigment Indices (RGI) | $R_{690}/R_{550}$ | B3, B5 | [40] |
| Anthocyanin Reflectance Index (ARI) | $(1/R_{green}) * (1/R_{red\,edge})$ | B3, B8a | |
| Carotenoid Reflectance (CRI) | $(R_{510})^{-1} - (R_{550})^{-1}$ | B2, B3 | [30] |
| Carotenoid Reflectance (CRI2) | $(R_{510})^{-1} - (R_{700})^{-1}$ | B2, B5 | |
| Visible Atmospherically Resistant Index (VARI$_{green}$) | $(R_{green} - R_{red})/(R_{green} + R_{red} - R_{blue})$ | B2, B3, B4 | |
| Cater Stress Index (CSI2) | $R_{695}/R_{760}$ | B5, B7 | [41] |
| Apparent Clumping Index (ACI) | $R_{green}/R_{NIR}$ | B3, B8 | [42] |
| Red-Edge Normalized Difference Vegetation Index (NDRE1) | $(R_{RE2} - R_{RE1})/(R_{RE2} + R_{RE1})$ | B5, B6 | [43] |
| Mangrove Index (MI) | $(R_{NIR} - R_{SWIR}/R_{NIR} * R_{SWIR}) * 10,000$ | B8, B12 | [44] |
| Mangrove Forest Index (MFI) | $[(R_{RE1} - R_{B\lambda 1}) + (R_{RE2} - R_{B\lambda 2}) + (R_{RE3} - R_{B\lambda 3}) + (R_{RE4} - R_{B\lambda 4})]/4$ $R_{B\lambda i} = R_{2190} + (R_{665} - R_{2190}) * (2190 - \lambda i)/(2190 - 665)$ | B4, B5, B6, B7, B8a, B12 | [45] |

*2.4. Estimation of Mangrove Seasonal Leaf Nutrients with Machine Learning Method*

2.4.1. Three Machine Learning Models

In this study, three machine learning models were employed to estimate mangrove leaf nutrients with seasonal Sentinel-2 images: XGBoost (extreme gradient boosting), random forest (RF), and light gradient boosting machine (LightGBM). All three methods are ensemble learning-based algorithms, which could improve the generalization ability and robustness of basic learners by combining the prediction results of multiple base learners.

XGBoost is proposed based on the gradient-boosting decision tree (GBDT) [46]. Compared to the traditional GBDT algorithm, XGBoost adopts second-order Taylor expansion into the cost function to avoid model overfitting [46]. Moreover, XGBoost uses a sparse-aware split lookup method to process sparse data [47], which is practical for dealing with the limited number of observation samples in the field of quantitative remote sensing [48]. In the training stage, one decision tree is incremented during each iteration, gradually forming a strong evaluator with a combination of multiple trees. The objective function for the *t*th iteration is defined as following:

$$obj^{(t)} = \sum_{i=1}^{n} \left[ g_i f_t(x_i) + \frac{1}{2} h_i f_t^2(x_i) \right] + \omega(f_t) \tag{1}$$

$$g_i = \partial_{\hat{y}^{(t-1)}} l\left(y_i, \hat{y}^{(t-1)}\right)$$
$$h_i = \partial_{\hat{y}^{(t-1)}}^2 l\left(y_i, \hat{y}^{(t-1)}\right) \tag{2}$$

where $y_i$ is the field-measured value; $\hat{y}_i^{(t-1)}$ is the predicted value of $(t-1)$-th iteration; $f_t$ represents the *k*th decision tree; $x_i$ represents the feature vector of the *i*th sample; $l\left(y_i, \hat{y}^{(t-1)}\right)$ is the prediction error of the learning model consisting of the previous $t-1$ trees; $g_i$ and $h_i$ are the first and second derivative of the prediction model and the current model, respectively; and $\omega(f_t)$ is the regular term of the objective function in each iteration.

RF is proposed based on the bagging method [49]; it establishes multiple decision trees and mutually independent weak estimators at a time, and it votes on all independent weak estimators. RF selects the optimal estimators with the highest votes as the final model-prediction result. The algorithm generates a new training sample set by randomly extracting *k* samples from the original training sample set, and then generates *k* decision trees according to the bootstrapping sample set to form a random forest. Several studies claimed that XGBoost was superior to the popular machine learning method (RF) in the remote estimation of the biomass of mangrove and the chlorophyll of pepper leaf [10,49].

LightGBM is proposed based on GBDT with a different splitting strategy of the leaf nodes [50]. Unlike the undifferentiated level-wise strategy of XGBoost, LightGBM uses a leaf-wise strategy of leaf node splitting which selects the nodes that benefit most from splitting and reduces computational effort [50]. The algorithm employs gradient-based one-side sampling (GOSS) and exclusive feature bundling (EFB) for faster training. GOSS picks data with a larger gradient from the samples to increase their contribution to the calculated information gain. EFB reduces the data dimensionality by merging similar data features. The objective function depending on the leaf-wise strategy is defined as follows:

$$G = \frac{1}{2} \left( \frac{\left(\sum_{i \in I_L} g_i\right)^2}{\sum_{i \in I_L} h_i + \lambda} + \frac{\left(\sum_{i \in I_R} g_i\right)^2}{\sum_{i \in I_R} h_i + \lambda} - \frac{\left(\sum_{i \in I} g_i\right)^2}{\sum_{i \in I} h_i + \lambda} \right) \tag{3}$$

where $g_i$ and $h_i$ indicate the first and second derivative statistics of the loss function, and $I_L$ and $I_R$ are the sample sets of the leaf and right branches, respectively.

All three methods have the efficient ability of feature selection based on the importance score, which can be quantified using several metrics, such as the gain, weights, total gain, and total coverage of a node. We chose the metric value of gain to explain the relative importance of each corresponding feature. The higher score values indicate a greater

contribution to the model performance. The three models were built using Python version 3.8, and the settings of the main parameters are shown in Table 5.

**Table 5.** Main parameter settings of the three machine learning methods.

| Algorithm | Algorithm Library | Main Parameter Settings |
|---|---|---|
| XGBoost | https://github.com/dmlc/xgboost/ (accessed on 20 September 2021), version 1.5.0 | max_depth = 5, learning_rate = 0.1, n_estimators = 200, min_child_weight = 1 |
| RF | https://github.com/kjw0612/awesome-random-forest (accessed on 20 September 2021), version 1.2.2 | n_estimators = 200, criterion = 'mse', max_depth = None, min_samples_split = 2, min_samples_leaf = 1 |
| LightGBM | https://github.com/microsoft/LightGBM (accessed on 20 September 2021), version 3.3.1 | learning_rate = 0.1, num_leaves = 31, min_data_in_leaf = 20, n_estimators = 200 |

### 2.4.2. Two Modeling Strategies

In this study, two modeling strategies were used to compare the performances of the three machine learning models across the three seasons: model development with the dataset of a single season and the pooled dataset of three seasons. Due to the possible data redundancy of the spectral features of Sentinel-2 images, feature selection was required to choose the sensitive features for modeling. Based on the importance score derived from each machine learning method, the features with an importance score greater than the standard deviation of all score values were selected for further modeling.

For the first modeling strategy, a total of nine regression models (three machine learning methods × three seasons) were established and cross-validated by the leave-one-out cross-validation procedure in estimating each leaf nutrient. For the second modeling strategy, all the samples were combined, a total of three regression models were established and cross-validated by estimating each leaf nutrient, and the samples in each season were predicted by the combined model.

### 2.4.3. Model Evaluation

The coefficient of determination ($R^2$), relative root mean square error (RRMSE), and residual prediction deviation (RPD) were calculated to evaluate the performance of each model:

$$R^2 = 1 - \frac{\sum_{i=1}^{n}(y_i - \hat{y}_i)^2}{\sum_{i=1}^{n}(y_i - \overline{y_i})^2} \tag{4}$$

$$RMSE = \sqrt{\frac{\sum_{i=1}^{n}(y_i - \hat{y}_i)^2}{n}} \tag{5}$$

$$RRMSE = RMSE/\overline{y_i} \tag{6}$$

$$RPD = SD/RMSE \tag{7}$$

$$MAE = \frac{1}{n} \times \sum_{i=1}^{n}\left(\frac{|y_i - \hat{y}_i|}{y_i} \times 100\%\right) \tag{8}$$

where $y_i$ and $\hat{y}_i$ are the measured and predicted value of the $i$th sample, respectively; $\overline{y_i}$ is the mean value of measured leaf nutrient, $n$ is the number of leaf samples, and SD is the standard deviation. A higher $R^2$ and RPD and a lower RRMSE (unit: %) indicate a better model performance.

Based on the best-performing model, all the combined samples of the three seasons were classified into four groups according to the quartile values of the leaf nutrients, and the number of samples and mean absolute error (MAE, unit: %) of each group were calculated.

*2.5. Mapping of Seasonal Mangrove Leaf Nutrients*

For each nutrient, the optimal machine learning model was chosen to map its spatial distribution in each season. To investigate the spatial distribution of leaf nutrients, hotspot analysis was further conducted in ArcGIS 10.8 using the Getis-Ord Gi* parameters to identify statistically significant clusters of hot and cold spots. The *p* value and *z* score were obtained to judge whether to reject the null hypothesis. The *p* value indicates the probability that the spatial pattern of the leaf nutrient concentration was created by a random process. The Z score indicates the multiples of the standard deviation, and a higher Z score (>0) with a lower *p* value indicates greater clustering of higher value (hot spot), while a lower Z score (<0) with a lower *p* value indicates greater clustering of lower value (cold spot). Moreover, there is no significant spatial clustering when the Z score is close to 0.

On the other hand, to understand the changes of seasonal leaf nutrients in mangrove forests across different years, we obtained another 12 cloudless Sentinel-2 images (Table 6) from 2017 to 2021 to map the concentrations of leaf C, N, and P based on the corresponding best-performing model.

**Table 6.** The acquisition time of 12 Sentinel-2 images from 2017 to 2021.

| Year | Season | Acquisition Date |
|------|--------|------------------|
| 2017 | Spring | 8 April 2017 |
|      | Summer | 27 June 2017 |
|      | Winter | 19 December 2017 |
| 2018 | Summer | 31 August 2018 |
|      | Winter | 23 January 2019 |
| 2019 | Spring | 23 May 2019 |
|      | Summer | 2 July 2019 |
| 2020 | Spring | 8 March 2020 |
|      | Winter | 8 December 2020 |
| 2021 | Spring | 17 May 2021 |
|      | Summer | 4 September 2021 |
|      | Winter | 28 November 2021 |

## 3. Results

*3.1. Seasonal Variation of Mangrove Leaf Nutrients*

Among the three seasons (Table 2), the leaf samples from summer showed higher leaf C concentration (mean = 492.75 g/kg) than those from spring (mean = 449.07 g/kg) and winter (mean = 457.16 g/kg). However, higher leaf N (mean = 14.94 g/kg) and P (mean = 1.58 g/kg) concentrations were observed in winter than in spring (mean = 10.58 and 0.90 g/kg) and summer (mean = 10.59 and 0.86 g/kg). Moreover, considering any two seasons, the one-way analysis of variation (ANOVA) results (Table 7) showed that there was no significant difference in leaf C between spring and winter ($p > 0.05$), and no significant difference in leaf N or P was observed between spring and summer. However, considering all the three seasons, there were significant differences in leaf C, N, and P ($p < 0.05$).

Based on all the leaf samples from the three seasons (Figure 2), leaf N was positively correlated with leaf P, while leaf C was negatively correlated with leaf N and P ($p < 0.01$), and the N–P correlation was stronger than the C–N and C–P correlation. Based on the leaf samples from a single season, the N–P correlation (0.880, 0.848, and 0.686 in winter, summer, and spring, respectively; $p < 0.01$) was also stronger than the C–P ($-0.716$, $-0.631$, and $-0.420$; $p < 0.01$) and C-N correlation ($-0.612$, $-0.526$, and $-0.334$; $p < 0.01$).

**Table 7.** ANOVA of leaf nutrients between seasons.

| Nutrient | Season | Spring | Summer | Winter |
|---|---|---|---|---|
| C | Spring | | | |
| | Summer | 0.000 * | | |
| | Winter | 0.078 | 0.000 * | |
| N | Spring | | | |
| | Summer | 0.978 | | |
| | Winter | 0.000 * | 0.000 * | |
| P | Spring | | | |
| | Summer | 0.586 | | |
| | Winter | 0.000 * | 0.000 * | |

* Significant ($p < 0.05$).

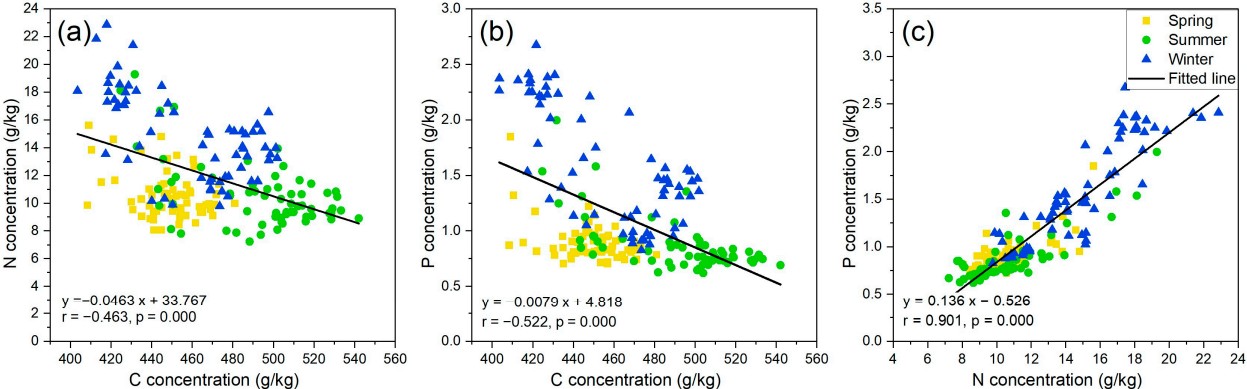

**Figure 2.** Intercorrelation of leaf C–N concentration (**a**), leaf C–P concentration (**b**), and leaf N–P concentration (**c**) with all the samples (*n* = 172), respectively.

### 3.2. Correlation of Leaf Nutrients against Spectral Features of Seasonal Sentinel-2 Images

Leaf C had the highest correlation with DSWI index, B11, and MCARI/OSAVI ($r$ = 0.693, 0.751, and 0.743, $p < 0.01$) in spring, summer, and winter, respectively (Figure 3); leaf N had the highest correlation with CIgreen, B11, and B6 ($r$ = −0.351, −0.450, and −0.633, $p < 0.01$) in spring, summer, and winter, respectively; leaf P had the highest correlation with B3, B11, and B6 ($r$ = 0.490, −0.561, and −0.730, $p < 0.01$) in spring, summer, and winter, respectively. Considering the mean absolute correlation coefficient of the 42 spectral features against leaf nutrients, the three leaf nutrients reported stronger correlation (mean $|r|$ = 0.51, 0.44, and 0.49 for C, N, and P, respectively) in winter than summer (mean $|r|$ = 0.50, 0.19, and 0.23) and spring (mean $|r|$ = 0.39, 0.28, and 0.36).

### 3.3. Comparison of Three Machine Learning Models in Estimating Leaf Nutrients

When using the dataset of a single season to estimate the three leaf nutrients (Table 8), XGBoost model with sensitive features (Figure 4) reported higher accuracy ($R^2$ = 0.655–0.829, RRMSE = 1.687–2.408%, RPD = 1.703–2.418 for leaf C estimation; $R^2$ = 0.668–0.743, RRMSE = 6.090–9.668%, RPD = 1.736–1.973 for leaf N estimation; $R^2$ = 0.539–0.622, RRMSE = 4.659–19.560%, RPD = 1.473–1.627 for leaf P estimation) than RF and LightGBM in each season, except the cases of leaf C estimation in summer and leaf P estimation in summer and winter. Considering the mean value of the performance parameters of the three machine learning models, leaf C was estimated with higher accuracy in winter (mean $R^2$ = 0.829, mean RRMSE = 2.401%, mean RPD = 2.418) than in summer and spring; leaf N and P were estimated with higher accuracy in summer (mean $R^2$ = 0.743 and 0.622, mean RRMSE = 9.668% and 16.251%, mean RPD = 1.973 and 1.627) than spring and winter.

When using the pooled dataset of three seasons to estimate leaf nutrients (Table 9), all three models reported very poor performance in spring ($R^2 < 0.3$), and the XGBoost model with sensitive spectral features (Figure 4) showed a stronger performance (mean $R^2$ = 0.513, 0.347, and 0.389 for C, N, and P estimation) than RF (mean $R^2$ = 0.477, 0.262,

and 0.376) and LightGBM (mean $R^2$ = 0.453, 0.232, and 0.33) considering the mean value of $R^2$ in summer and winter. Considering the mean value of the performance parameters of the three models, leaf C was estimated with higher accuracy in summer (mean $R^2$ = 0.788, mean RRMSE = 1.604%, mean RPD = 2.171) than in winter and spring; leaf N and P were estimated with higher accuracy in winter (mean $R^2$ = 0.504 and 0.439, mean RRMSE = 9.321% and 12.993%, mean RPD = 1.419 and 1.335) than in summer and spring.

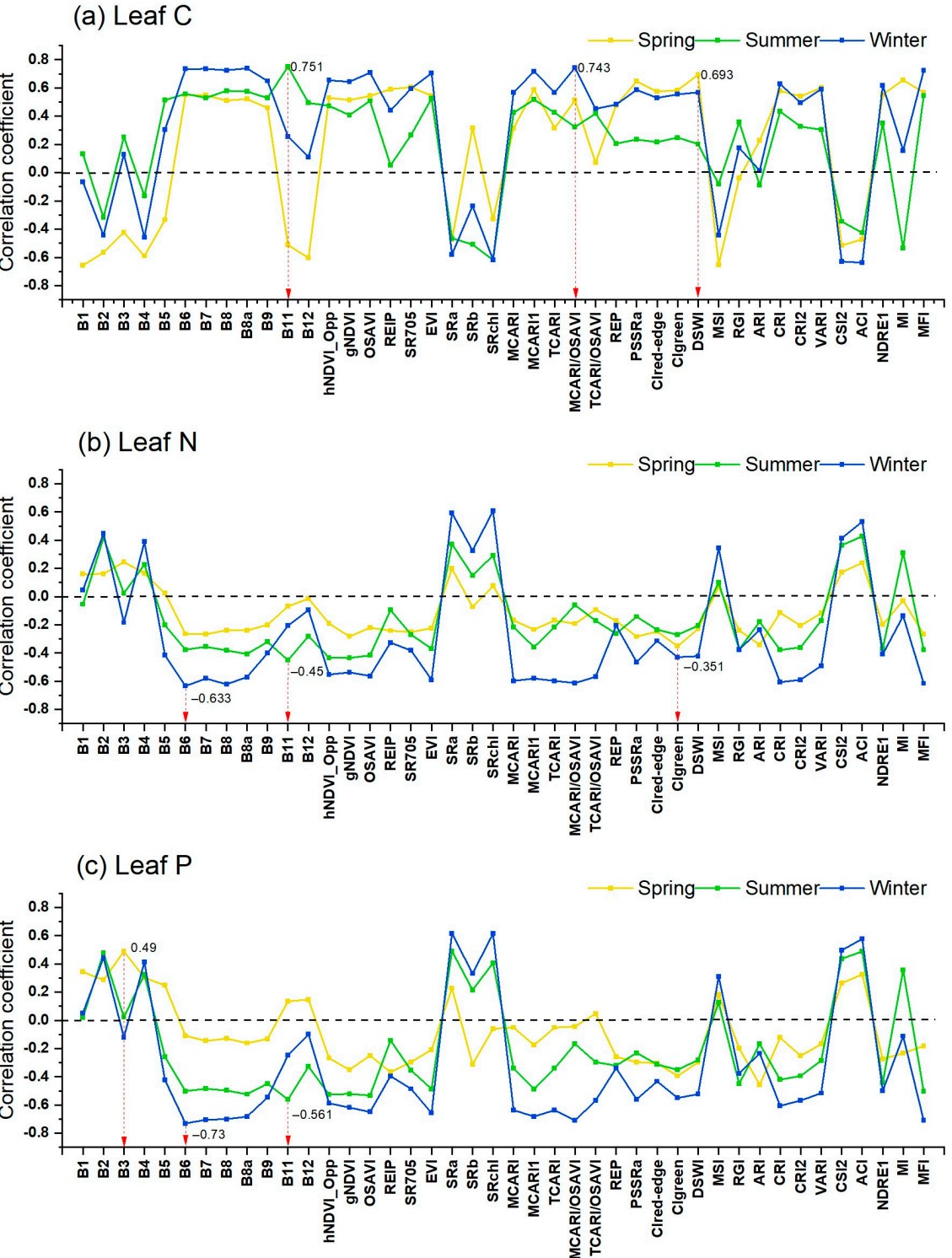

**Figure 3.** Correlations of leaf C (**a**), N (**b**), and P (**c**) against spectral features (12 spectral bands + 30 VIs) of Sentinel-2 images in spring, summer, and winter. The red arrow indicates the feature with the highest correlation in each season.

**Table 8.** Model performance in estimating leaf C, N, and P using the dataset of a single season.

| Model | Nutrient | Spring | | | Summer | | | Winter | | |
|---|---|---|---|---|---|---|---|---|---|---|
| | | $R^2$ | RRMSE(%) | RPD | $R^2$ | RRMSE(%) | RPD | $R^2$ | RRMSE(%) | RPD |
| XGBoost | C | 0.655 | 1.687 | 1.703 | 0.799 | 2.408 | 2.230 | 0.829 | 2.401 | 2.418 |
| | N | 0.668 | 6.090 | 1.736 | 0.743 | 9.668 | 1.973 | 0.704 | 8.998 | 1.838 |
| | P | 0.539 | 4.659 | 1.473 | 0.622 | 16.251 | 1.627 | 0.596 | 19.560 | 1.573 |
| RF | C | 0.549 | 1.717 | 1.489 | 0.811 | 2.314 | 2.300 | 0.824 | 2.420 | 2.383 |
| | N | 0.629 | 13.699 | 1.642 | 0.684 | 10.063 | 1.779 | 0.637 | 9.944 | 1.660 |
| | P | 0.415 | 6.168 | 1.309 | 0.652 | 14.662 | 1.700 | 0.613 | 18.156 | 1.607 |
| LightGBM | C | 0.415 | 1.952 | 1.308 | 0.401 | 3.014 | 1.292 | 0.803 | 2.538 | 2.253 |
| | N | 0.654 | 5.512 | 1.700 | 0.133 | 9.813 | 1.074 | 0.015 | 3.835 | 1.008 |
| | P | 0.273 | 6.755 | 1.173 | 0.207 | 14.292 | 1.122 | 0.627 | 17.596 | 1.637 |

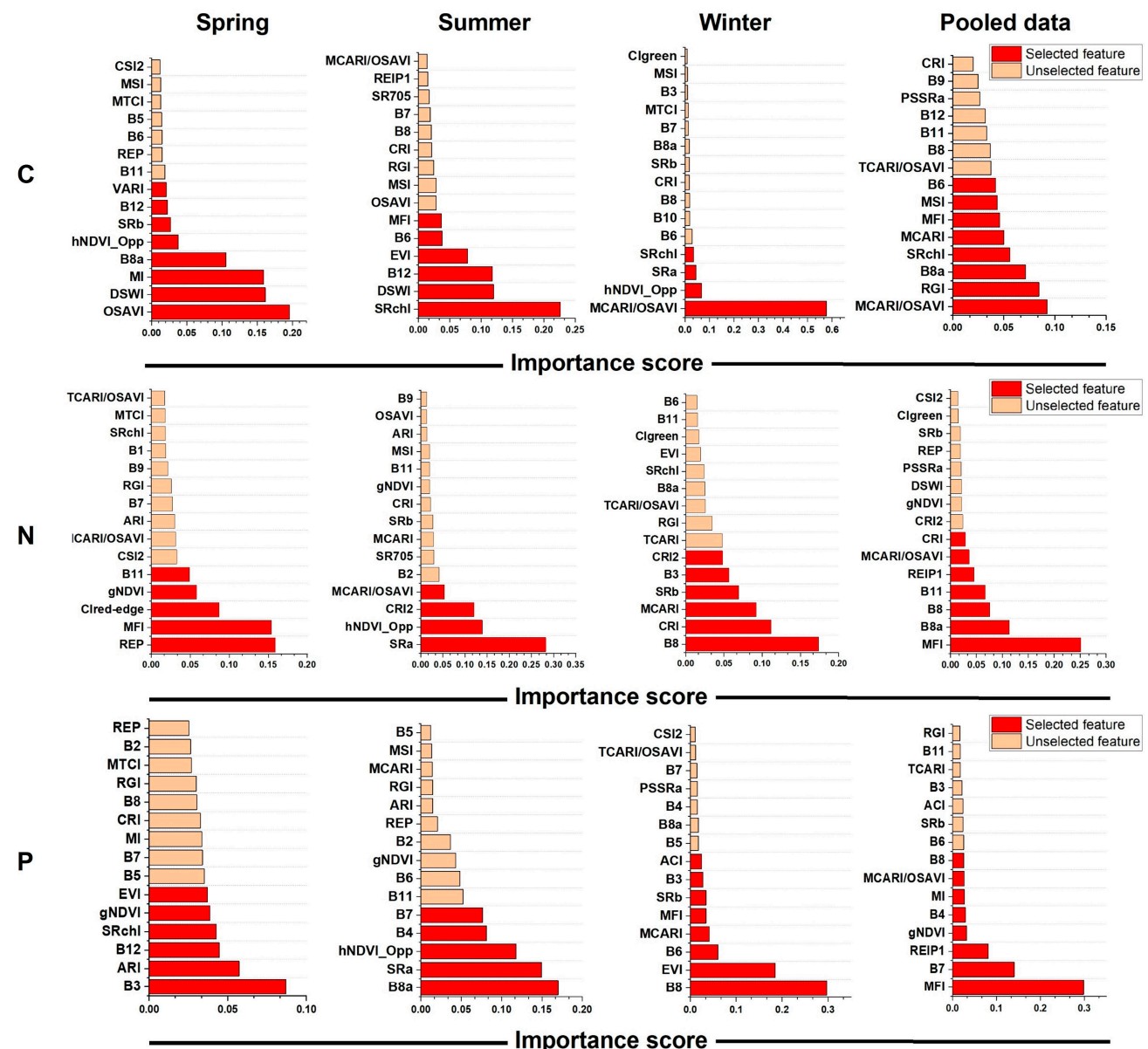

**Figure 4.** The ranking of the importance scores of the spectral features (bands + VIs) of Sentinel-2 images in correlating with leaf C, N, and P using the dataset of a single season and the pooled dataset of the three seasons. The importance scores were derived from the XGBoost algorithm, and the selected features were marked in red.

**Table 9.** Model performance in estimating leaf C, N, and P using the pooled dataset of three seasons.

| Model | Nutrient | Spring | | | Summer | | | Winter | | |
|---|---|---|---|---|---|---|---|---|---|---|
| | | $R^2$ | RRMSE(%) | RPD | $R^2$ | RRMSE(%) | RPD | $R^2$ | RRMSE(%) | RPD |
| XGBoost | C | 0.218 | 1.842 | 1.131 | 0.788 | 1.604 | 2.171 | 0.534 | 2.599 | 1.464 |
| | N | 0.021 | 9.757 | 1.011 | 0.504 | 9.321 | 1.419 | 0.516 | 10.481 | 1.438 |
| | P | 0.057 | 15.218 | 1.030 | 0.434 | 13.635 | 1.329 | 0.677 | 14.180 | 1.759 |
| RF | C | 0.133 | 1.987 | 1.074 | 0.730 | 1.818 | 1.924 | 0.569 | 2.543 | 1.523 |
| | N | 0.038 | 11.251 | 1.020 | 0.294 | 11.894 | 1.190 | 0.453 | 9.933 | 1.352 |
| | P | 0.079 | 16.392 | 1.042 | 0.439 | 12.993 | 1.335 | 0.611 | 15.016 | 1.603 |
| LightGBM | C | 0.169 | 2.094 | 1.097 | 0.758 | 1.714 | 2.035 | 0.432 | 2.882 | 1.327 |
| | N | 0.000 | 12.883 | 1.000 | 0.405 | 10.167 | 1.297 | 0.290 | 12.965 | 1.186 |
| | P | 0.000 | 19.315 | 1.000 | 0.417 | 15.544 | 1.309 | 0.573 | 16.101 | 1.530 |

Overall, the three models performed better using the dataset of a single season than using the pooled dataset of three seasons in estimating the three leaf nutrients in each season. Moreover, the XGBoost model always provided higher accuracy than RF and LightGBM in estimating the three leaf nutrients. Hence, an XGBoost model with sensitive spectral features using a single dataset was further used for mapping the spatial distribution of seasonal leaf nutrients from 2017 to 2021.

Based on the XGBoost method (Figure 4), MCARI/OSAVI using the bands of B3, B4, B5, and B8; OSAVI using the bands of B4 and B8, and SRchl using the bands of B3, B5, and B8 were the most sensitive to leaf C; REP using the bands of B4, B5, B6, and B7, SRa using the bands of B3, B4, B5, and B8, and MFI using the bands of B4, B5, B6, B7, B8a, and B12 were the most sensitive to leaf N; and B3, B8a, B8, and MFI were the most sensitive to leaf P.

On the other hand, the scatter plots of field-measured versus estimated leaf nutrients (Figure 5) showed that the estimated concentrations of leaf nutrients in spring were lower than in summer and winter. Due to the absence of samples with measured leaf C concentrations of 460–470 g/kg, two-point clusters of leaf C with abnormal distribution were observed in summer and winter. Moreover, the samples with medium C concentrations (443.63–473.10 g/kg) and lower N (8.52–11.72 g/kg) and lower P concentrations (0.71–1.12 g/kg) tended to have lower estimation errors (MAE < 10%, Table 10), while the samples with lower C concentrations (414.16–443.63 g/kg) and medium N (14.92–18.12 g/kg) and medium P concentrations (1.54–1.95 g/kg) tended to have higher estimation errors.

**Table 10.** The mean absolute error (MAE) of measured versus estimated leaf nutrients in different data groups based on the scatter points of Figure 5.

| Leaf Nutrient | Data Group | Spring | | Summer | | Winter | |
|---|---|---|---|---|---|---|---|
| | | *n* | MAE(%) | *n* | MAE(%) | *n* | MAE(%) |
| C | 414.16–443.63 | 14 | 1.98 | 4 | 3.06 | 24 | 2.37 |
| | 443.63–473.10 | 39 | 1.28 | 7 | 1.59 | 10 | 2.07 |
| | 473.10–502.57 | 0 | – | 22 | 2.52 | 28 | 1.98 |
| | 502.57–532.04 | 0 | – | 24 | 1.84 | 0 | – |
| N | 8.52–11.72 | 46 | 6.68 | 50 | 8.55 | 7 | 6.87 |
| | 11.72–14.92 | 7 | 5.53 | 3 | 9.53 | 22 | 6.30 |
| | 14.92–18.12 | 0 | – | 4 | 12.22 | 27 | 9.14 |
| | 18.12–21.33 | 0 | – | 0 | – | 6 | 7.81 |
| P | 0.71–1.12 | 53 | 7.62 | 49 | 9.50 | 14 | 9.02 |
| | 1.12–1.54 | 0 | – | 6 | 21.41 | 19 | 22.09 |
| | 1.54–1.95 | 0 | – | 2 | 25.52 | 7 | 22.59 |
| | 1.95–2.37 | 0 | – | 0 | – | 22 | 15.36 |

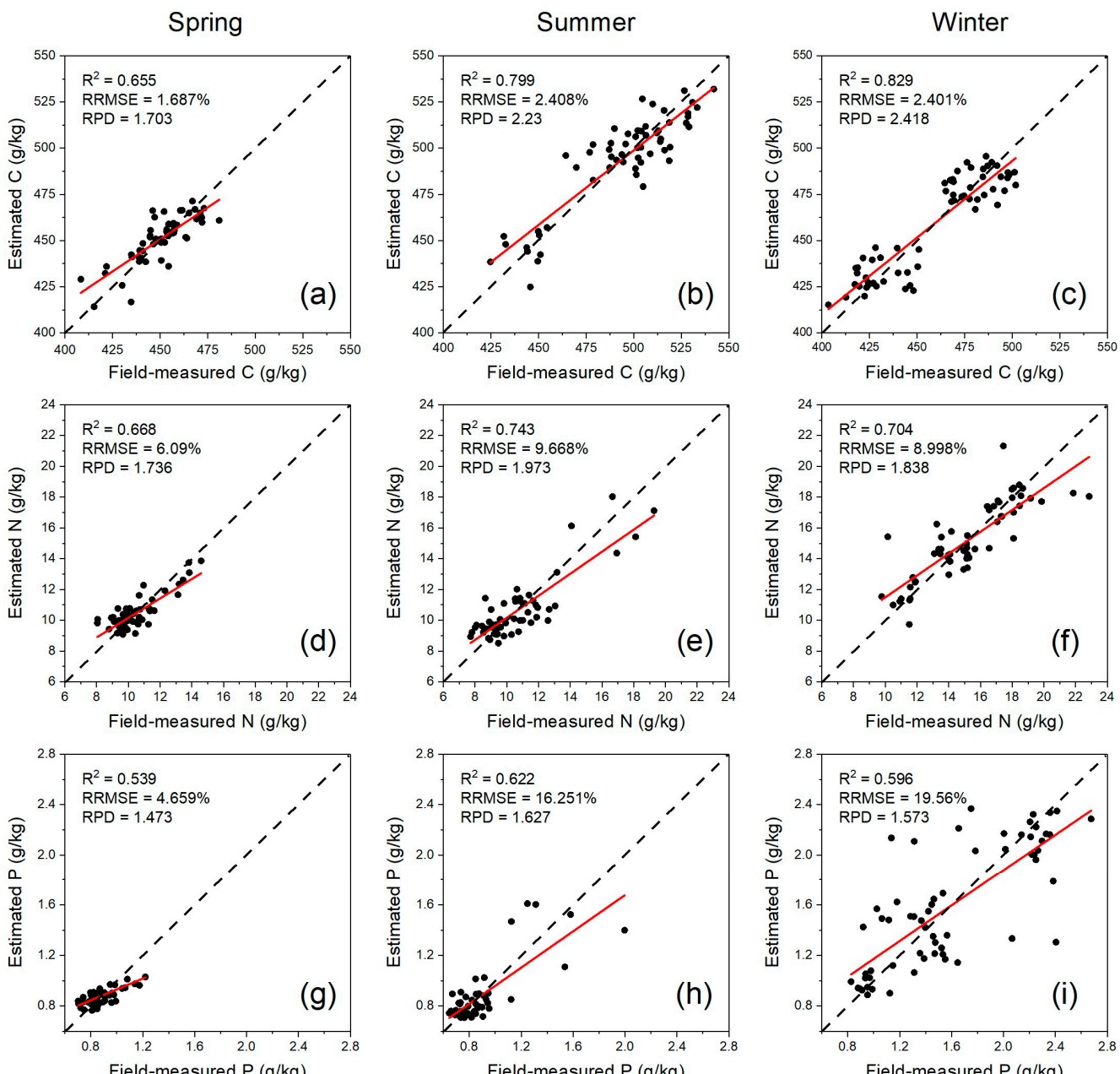

**Figure 5.** Scatter plots of field-measured versus estimated leaf C (**a–c**), N (**d–f**), and P (**g–i**) using the XGBoost model with the dataset of a single season (the fitted line showed in red).

### 3.4. Mapping Seasonal Leaf Nutrients with XGBoost Model

#### 3.4.1. Mapping Leaf C, N, and P Concentrations in Three Seasons

Based on the XGBoost model using the sensitive spectral features of the dataset of a single season, leaf C, N, and P concentrations were mapped in spring, summer, and winter, respectively (Figure 6). The mapped C concentrations in summer (mean = 499.564 g/kg, range = 454.886–530.845 g/kg) were higher than those in winter (mean = 481.287 g/kg, range = 435.731–510.978 g/kg) and spring (mean = 449.757 g/kg, range = 406.960–481.936 g/kg). Leaf C concentrations were higher in the northwestern part of the study area, while lower C concentrations were mostly observed in the estuary and nearshore areas with low vegetation coverage.

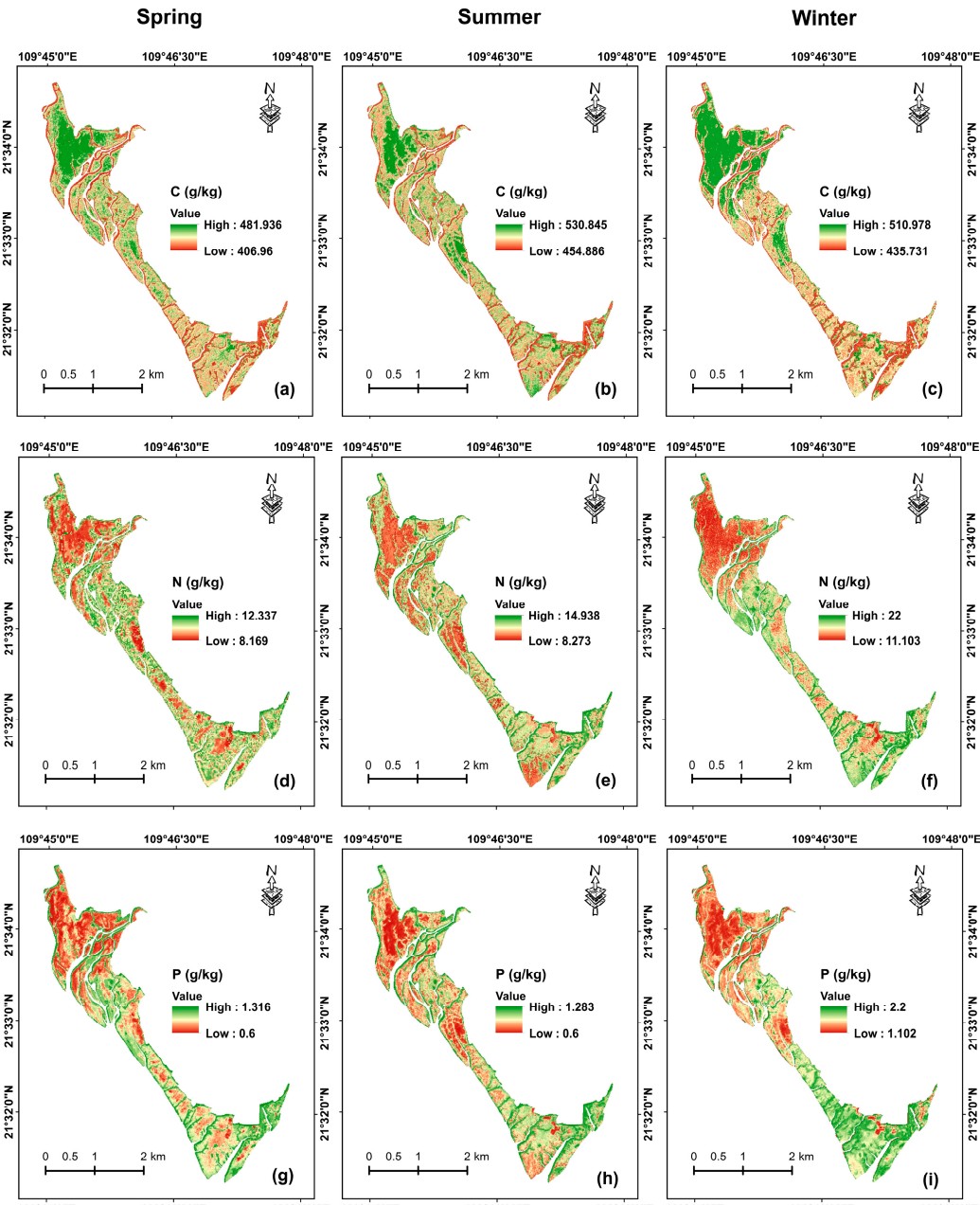

**Figure 6.** Maps of leaf C (**a**–**c**), N (**d**–**f**), and P (**g**–**i**) concentrations in spring, summer, and winter derived from the XGBoost model using sensitive spectral features of Sentinel-2 images.

The mapped N concentrations (mean = 16.937 g/kg, range = 11.103–22.000 g/kg) in winter were higher than those in spring (mean = 9.184 g/kg, range = 8.169–12.337 g/kg) and summer (mean = 9.059 g/kg, range = 8.273–14.938 g/kg). Moreover, the area with higher C was found to have lower N. The mapped leaf P concentrations showed similar spatial distribution characteristics to the mapped N concentrations, and leaf P showed higher concentrations in winter (mean = 1.513 g/kg, range = 1.102–2.200 g/kg) than in spring (mean = 0.832 g/kg, range = 0.600–1.316 g/kg) and summer (mean = 0.773, range = 0.600–1.283 g/kg).

The hotspot analysis of mapped leaf nutrients (Figure 7) showed that there were different spatial patterns of aggregation across different seasons. The significant hot spot areas of leaf C were mainly located in the northwest and middle part of the study area, while the significant cold spot areas of leaf C were mainly located near the rivers. In contrast, the aggregations of leaf N and P in most of the study area were not significant; the significant hot spot areas of leaf N and P were mainly located near the rivers and the

boundary, while the significant cold spot areas were mainly located in the northwest of the study area.

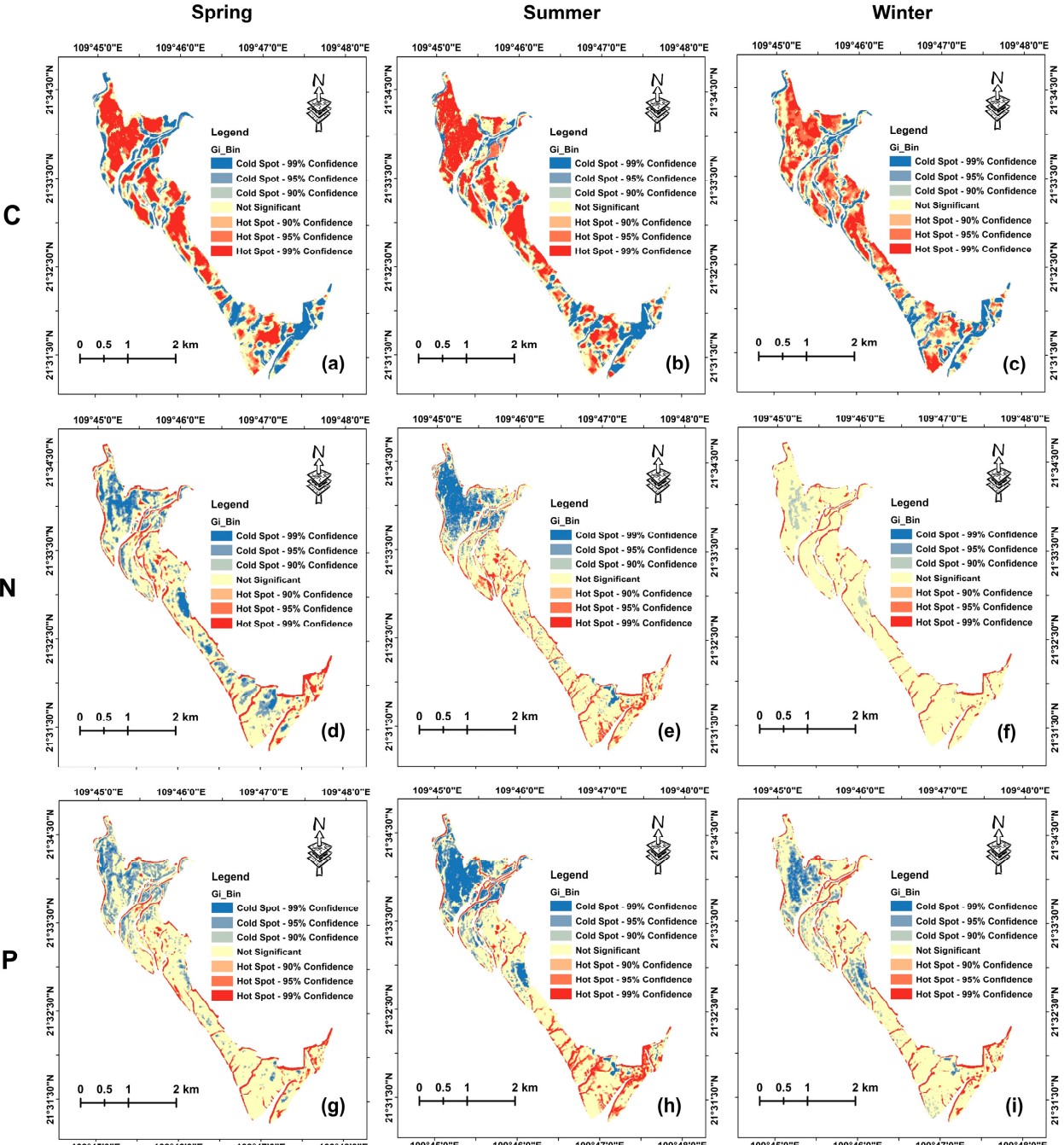

**Figure 7.** Hotspot analysis of mapped leaf C (**a**–**c**), N (**d**–**f**), and P (**g**–**i**) concentrations in spring, summer, and winter.

### 3.4.2. Mapping Seasonal Leaf C, N and P Concentrations from 2017 to 2021

During 2017–2021, the mean mapped leaf C concentration in summer (range = 463.003–499.564 g/kg) was higher with wider data variation than in winter (range = 459.631–481.287 g/kg) and in spring (range = 449.757–465.282 g/kg) (Figure 8); the mean mapped leaf N and P concentrations in winter were much higher than those in spring and summer (Figures 9 and 10). Moreover, the mean leaf N and P concentrations were stable with minor variations across the 15 seasons of the five years.

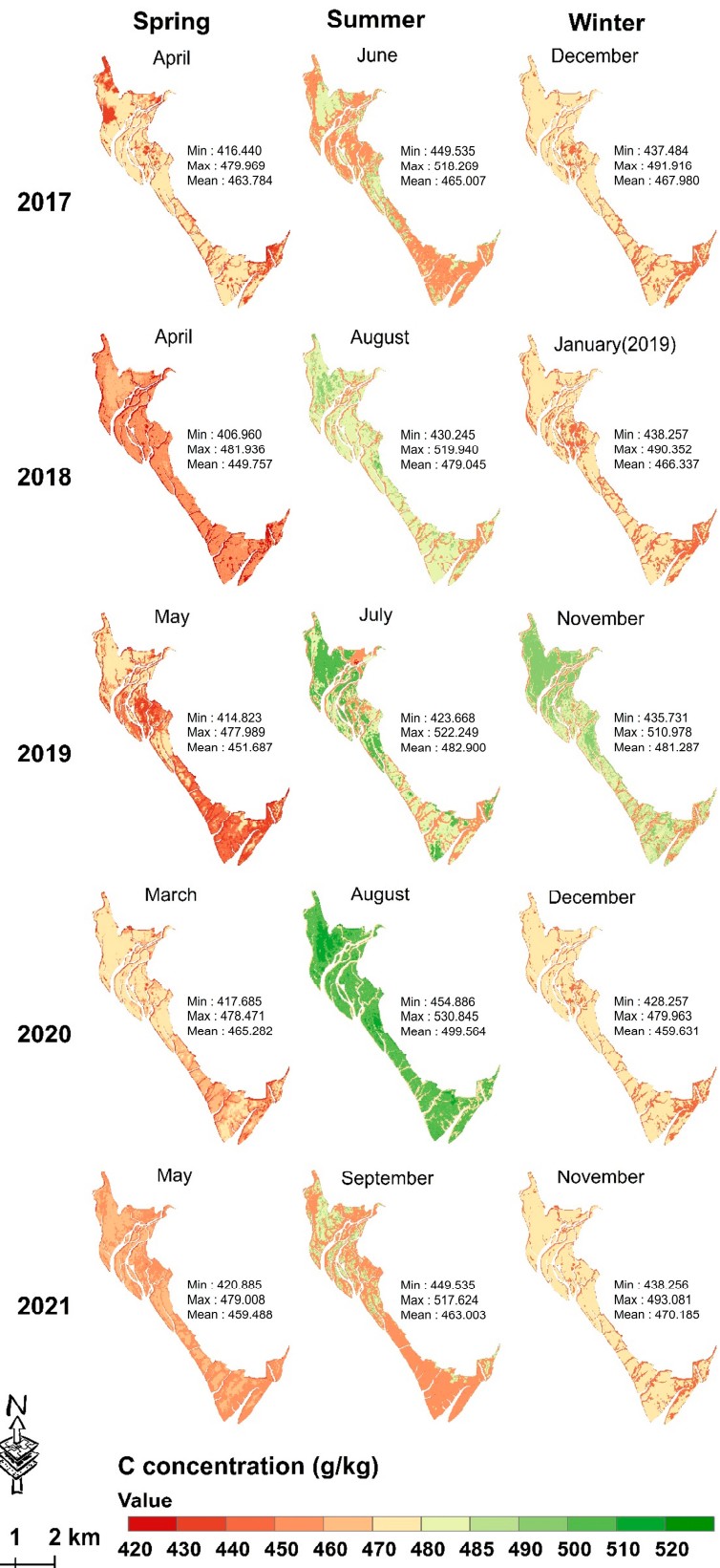

**Figure 8.** Seasonal leaf C mapping from 2017 to 2021 using the XGBoost model with the dataset of a single season.

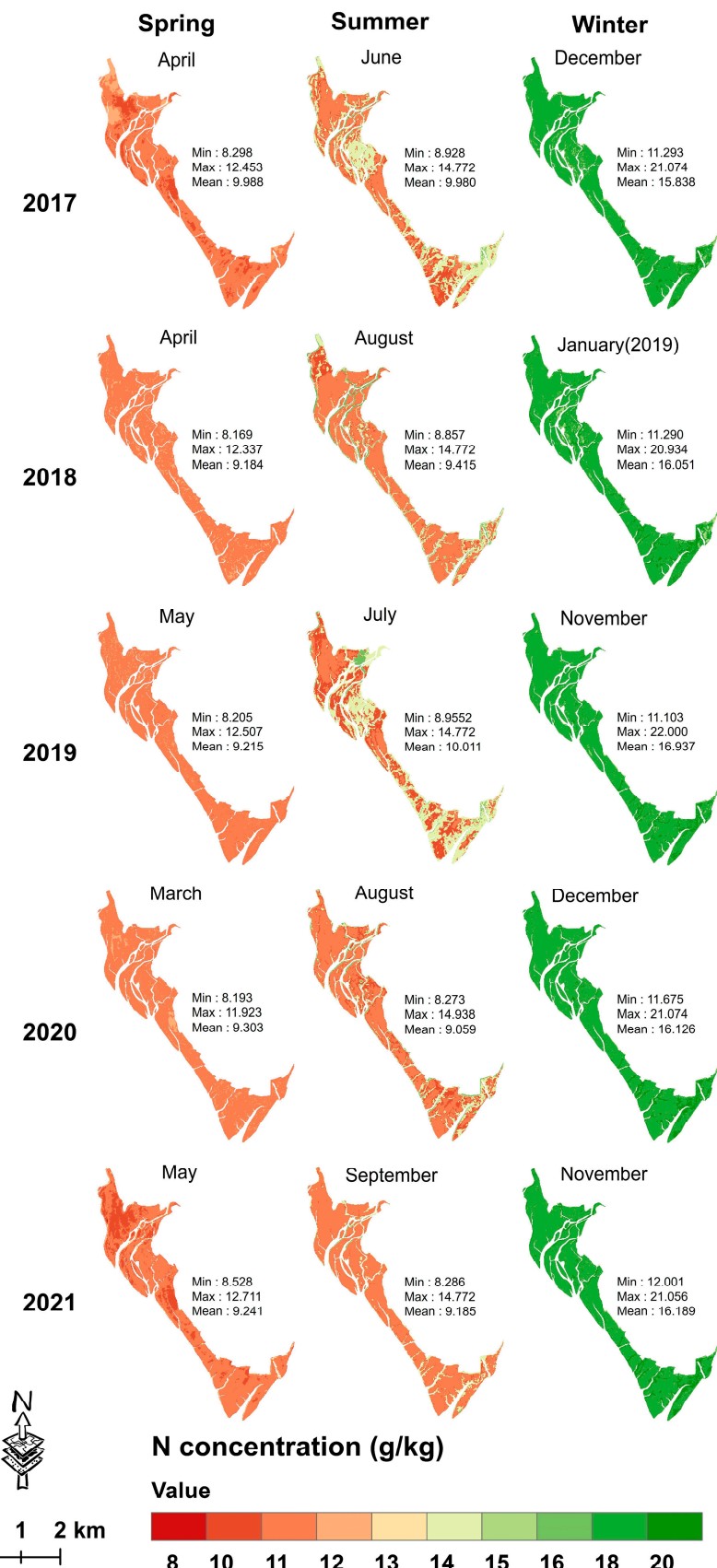

**Figure 9.** Seasonal leaf N mapping from 2017 to 2021 using the XGBoost model with the dataset of a single season.

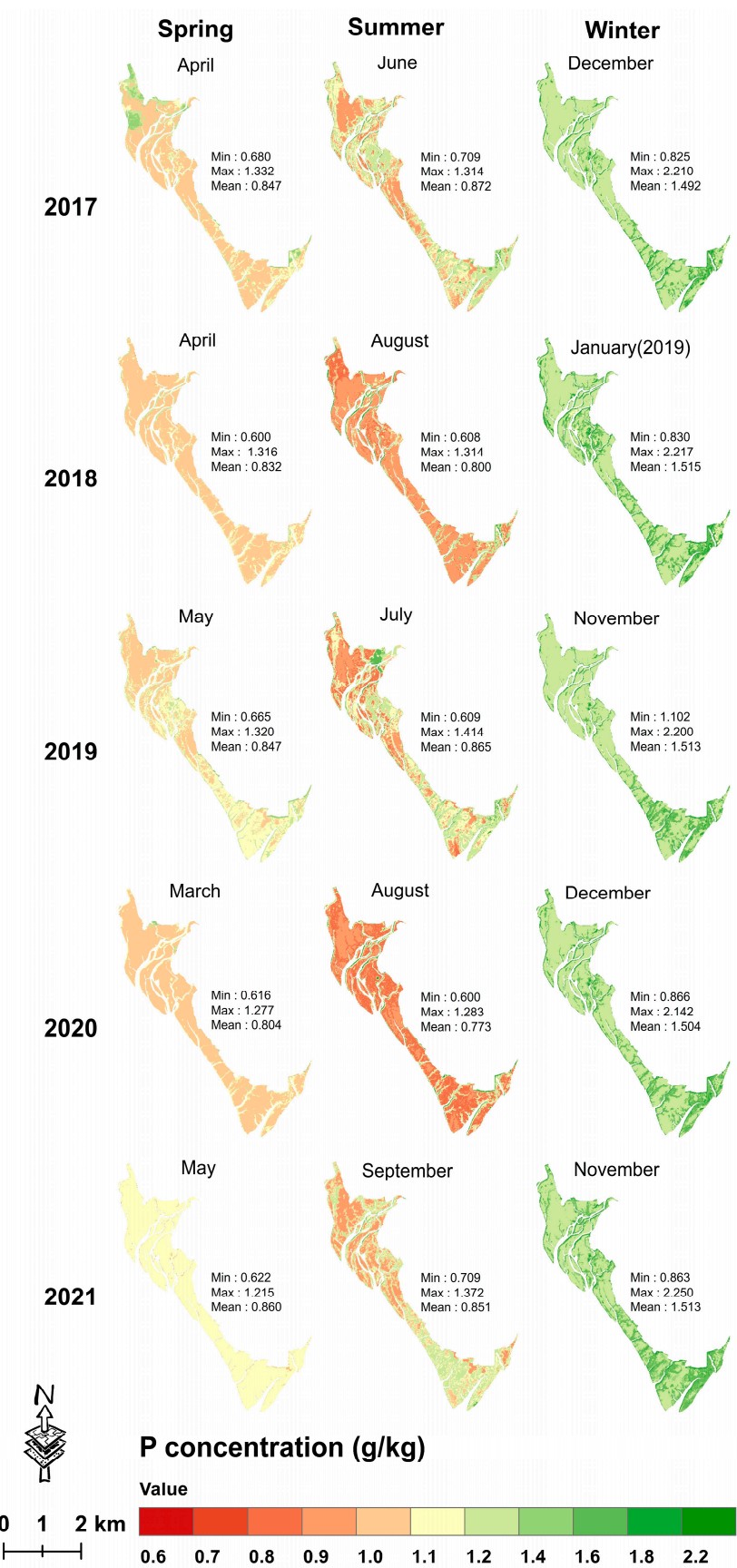

**Figure 10.** Seasonal leaf P mapping from 2017 to 2021 using the XGBoost model with the dataset of a single season.

## 4. Discussion

### 4.1. The Stoichiometry of Mangrove Leaf Nutrients across Different Seasons

The mean field-measured mangrove leaf C concentration of three seasons in Gaoqiao Mangrove Reserve was 466.33 g/kg, which is slightly higher than the globally averaged value of 492 terrestrial woody plants (464 g/kg) [51] and 11.53% higher than that of global coastal wetland plants (418.1 g/kg) [52]. The mean field-measured leaf N (12.13 g/kg) and P (1.13 g/kg) are 8.13–30.29% lower than the mean value of global coastal wetland vegetation (N = 16.1 g/kg, P = 1.6 g/kg) [52] and global terrestrial plants (N = 17.4 g/kg, P = 1.23 g/kg) [51]. These differences may be related to species types, phenology, and/or sampling strategies [53]. Such a comparison further confirms the fact that mangrove ecosystems have a strong C sequestration ability [54]. Moreover, the lower N/P ratios (<14) of the three seasons indicate that mangrove growth might be limited by N [55].

The seasonal trends of mapped leaf N and P concentration were similar (Figure 6) due to the strong correlation between them (Figure 3). Moreover, seasonality significantly affected the concentration of leaf N and P (Table 7), which agrees with the findings of Qin et al. [56]. However, Milla et al. [57] claimed that there was no significant correlation between the leaf N and P of woody plants, and Liu et al. [58] found that there was no strong seasonal variation in the leaf nutrients of *S. salsa* in the Yellow River Delta wetland. These results suggest that the nutrient utilization patterns could be affected by a variety of factors, such as climatic conditions, altitude, tidal levels, soil components, and species composition [59,60].

We found that leaf C was negatively correlated with N and P, and N was strongly correlated with P, which is in agreement with the findings of Michaels [61]. Such correlations also reflect the N and P utilization strategies in the C fixation process [62]. The fixation of C in the plant metabolism requires the participation of proteases (N storage), and the assembly of proteases requires the replication of nucleic acids (P storage) [61].

### 4.2. Sensitive Features Related to Mangrove Leaf Nutrients

The importance score ranking results (Figure 4) demonstrated that the red-edge bands (B5, B6, and B7, 705–783 nm) and near-infrared bands (B8 and B8a, 842–865 nm) performed better than other Sentinel-2 bands in estimating mangrove leaf nutrients, and the most sensitive VIs to leaf nutrients were mainly constructed by these bands. Moreover, B6 (740 nm) showed a higher correlation with the three nutrients than B5 (705 nm) and B7 (783 nm) (Figure 3), suggesting the superiority of B6 in correlating with mangrove leaf nutrients in different seasons, which agrees with the findings of Zhang et al. [63]. Many studies also demonstrated that red-edge bands are sensitive to leaf N and P across various plant species [13]. Moreover, the near-infrared bands are always used for developing NDVI, and the bands are less susceptible to saturation at high LAI and insensitive to unhealthy vegetation [12].

The simple ratio index (SRa, SRb, and SRchl) constructed by the ratio of two or three bands from B3, B4, B5, and B8 played an important role in estimating the three nutrients (Figure 4). Moreover, MCARI and its ratio with OSAVI were also sensitive to leaf C and N, and they have been widely used in the estimation of leaf chlorophyll due to the effective resistance to background interference and sensitivity to LAI saturation [34]. Several studies claimed that leaf chlorophyll is strongly correlated with leaf N [64], and leaf N and P had a close correlation, suggesting that chlorophyll content might be closely related to leaf C, N, and P estimation.

### 4.3. The Advantage of XGBoost in Estimating Mangrove Leaf Nutrients

Among the three machine learning models, in most cases, the XGBoost model was found to be optimal in estimating seasonal leaf nutrients using two modeling strategies (Tables 8 and 9). To our knowledge, this study was the first to estimate leaf C, N, and P in a mangrove forest using the XGBoost method and seasonal Sentinel-2 images. According to the interpretation of RPD [65], the XGBoost model using the dataset of a single season had

approximate quantitative estimations (RPD = 2.0–2.5) of leaf C in summer and winter and reported the possibility of distinguishing between high and low values of leaf C in spring and leaf N and P in three seasons (RPD = 1.5–2.0).

In most cases, it is difficult to obtain many sampling plots (15 m × 15 m) in mangrove forests due to the rough field accessibility, leading to a limited data range of leaf nutrients and underestimation of the concentrations of leaf nutrients to some extent. Hence, inevitable sparse sampling and the relatively weak spectral information of leaf N and P with relatively low concentrations are the greatest challenges to the accurate mapping of leaf nutrients in mangrove forests. Compared to other machine learning algorithms (e.g., RF), XGBoost uses a sparse-aware split lookup method to process sparse data, which is practical when dealing with the sparse sampling in a mangrove forest. Our results agree with the findings of Tian et al. [20] and Mohammadi et al. [66], who also claimed that XGBoost outperformed RF and LightGBM in estimating grapevine leaf N and hydrogen solubility in hydrocarbons. The reason might be that XGBoost is more capable of solving the problems of feature selection, overfitting, hyperparameter tuning, and local optimality [67].

### 4.4. Limitation of Leaf Nutrients Estimation with Seasonal Sentinel-2 Images

We found that the model performance in estimating leaf nutrients was weaker in spring than in summer and winter, and the model performance was extremely poor in spring with the pooled dataset of three seasons (Table 9). One possible reason might be that the field sampling in April 2018 covered a smaller portion of the study area (Figure 1) with a narrower data range of leaf nutrients (Figures 5 and 6). Hence, it is necessary to improve the field sampling strategy with more sampling plots and larger sampling areas to increase the model performance and transferability.

Though Sentinel-2 images could provide a convenient way to monitor seasonal leaf nutrients, the 10 m pixels of Sentinel-2 images largely contain more than one species, and a pixel with low coverage or nearshore mangroves is also affected by sediment and seawater, which could influence the spectral features of mangroves and further lead to errors in the estimation of leaf nutrients. We mapped the leaf nutrients of 15 seasons from 2017 to 2021; however, the extended XGBoost model was developed by only one image of a single season, which might lead to a lack of field validation of the mapped results for other years.

## 5. Conclusions

We compared three machine learning models to estimate mangrove leaf C, N, and P with Sentinel-2 images in spring, summer, and winter, and the best-performing model was extended to map leaf nutrients of 15 seasons from 2017 to 2021. The main conclusions could be drawn as follows:

(1) The XGBoost method had great potential for accurate estimation of mangrove leaf nutrients with seasonal Sentinel-2 images.
(2) Among the three nutrients, leaf C concentrations were the most accurately estimated, followed by leaf N and P.
(3) Red-edge (especially B6) and near-infrared bands (B8 and B8a) of Sentinel-2 images were efficient estimators of mangrove leaf nutrients.

Long-time seasonal monitoring of leaf nutrients could facilitate an understanding of the dynamic variation of C fixation, nutrient utilization, and growth status of mangrove forests. To achieve efficient monitoring with time series Sentinel-2 images, it is necessary to establish an inversion model of leaf nutrients with high accuracy and strong transferability. In future work, the species composition, elevation, LAI, and canopy height with more sampling plots in wider areas will be incorporated into the present model to improve model accuracy and applicability.

**Author Contributions:** Conceptualization and methodology, J.W. and J.M.; validation, J.W., J.M. and J.Z.; formal analysis, J.W., J.M. and J.Z.; investigation, J.W.; resources, J.W., J.M., J.Z., D.Z., X.J., Z.S., C.G. and G.W.; writing—original draft preparation, J.M.; writing—review and editing, J.W.;

supervision, G.W.; project administration, J.W.; funding acquisition, J.W. All authors have read and agreed to the published version of the manuscript.

**Funding:** This research was funded by the Guangdong Basic and Applied Basic Research Foundation (No. 2019A1515010741) and the Shenzhen Science and Technology Program (No. JCYJ20210324093210029).

**Data Availability Statement:** Not applicable.

**Conflicts of Interest:** The authors declare no conflict of interest. The funders had no role in the design of the study; in the collection, analyses, or interpretation of data; in the writing of the manuscript; or in the decision to publish the results.

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
