# Peer review of "Mapping Seasonal Leaf Nutrients of Mangrove with Sentinel-2 Images and XGBoost Method"

_remotesensing, doi:10.3390/rs14153679_

Round 1

Reviewer 1 Report

Dear authors,

Upon a careful reading and evaluation of your manuscript, I’m recommending it for a minor revision. This is an interesting topic, but I detect some deficiencies in its sections. In this regard, I have detailed my suggestions, which I believe will improve the overall quality of the manuscript content. I hope these comments are useful to you.

1. There needs to be a better description of why it is important to combine the XGBoost algorithm with Sentinel 2 images to map mangrove leaf nutrients. Just because these two models haven't been evaluated before in this task is a shallow justification. What are the real challenges faced by this type of mapping approach and why these algorithms may be useful to deal with it that others don't?

2. There is a text problem with lines 118, 119, and 120.

3. Please provide more details regarding the framework of the algorithm's implementation. What libraries were used? What parameters were chosen?

4. There's a citation error in lines 190-191.

5. In a general sense of this manuscript, the English language needs to be improved; there are grammar and sentence structure errors in the text. I advise a careful examination in a subsequent read.

Reviewer 2 Report

The paper "Mapping seasonal leaf nutrients of mangrove with Sentinel-2 images and XGBoost method" presents the ML based method estimating N/P/K concentration in mangrove leaves. The methodology and statistics are sounds. However, it has too huge figure and tables in the manuscript and sometimes ground observation work and remote sensing work are not consistent. (e.g., ground observation data to evaluate temporal dynamics was monitored considering the diferent of species. but could not found species difference detection by the sentinel-2 data and temporal analysis with remote sensing data was conducted just as regional average  ). The reviewer recommends the authors to devide this paper contents to 2 papers to simplify so that reader can smoothly understand the content without misleading. Specific comments are writen in the attached file.

Round 2

Reviewer 2 Report

The authors have reasonably addressed my concerns.